# The chimeric TAC receptor co-opts the T cell receptor yielding robust anti-tumor activity without toxicity

Christopher W. Helsen[1], Joanne A. Hammill[1], Vivian W.C. Lau [1], Kenneth A. Mwawasi[1], Arya Afsahi [1], Ksenia Bezverbnaya[1], Lisa Newhook[1], Danielle L. Hayes[1], Craig Aarts[1], Bojana Bojovic [1], Galina F. Denisova[1], Jacek M. Kwiecien[1,2], Ian Brain[1], Heather Derocher[3], Katy Milne[3], Brad H. Nelson[3] & Jonathan L. Bramson [1]

Engineering T cells with chimeric antigen receptors (CARs) is an effective method for directing T cells to attack tumors, but may cause adverse side effects such as the potentially lethal cytokine release syndrome. Here the authors show that the T cell antigen coupler (TAC), a chimeric receptor that co-opts the endogenous TCR, induces more efficient anti-tumor responses and reduced toxicity when compared with past-generation CARs. TAC-engineered T cells induce robust and antigen-specific cytokine production and cytotoxicity in vitro, and strong anti-tumor activity in a variety of xenograft models including solid and liquid tumors. In a solid tumor model, TAC-T cells outperform CD28-based CAR-T cells with increased anti-tumor efficacy, reduced toxicity, and faster tumor infiltration. Intratumoral TAC-T cells are enriched for Ki-67$^+$ CD8$^+$ T cells, demonstrating local expansion. These results indicate that TAC-T cells may have a superior therapeutic index relative to CAR-T cells.

[1] Department of Pathology and Molecular Medicine, McMaster University, 1280 Main St W, Hamilton, ON L8S 4L8, Canada. [2] Department of Clinical Pathomorphology, Medical University of Lublin, Racławickie 1 Street, 20-059 Lublin, Poland. [3] Trev & Joyce Deeley Research Centre, British Columbia Cancer Agency, 2410 Lee Ave, Victoria, BC V8R 6V5, Canada. Correspondence and requests for materials should be addressed to J.L.B. (email: bramsonj@mcmaster.ca)

Adoptive T cell transfer (ACT) involves the ex vivo expansion of a patient's T cells followed by infusion of the cell product into the patient. ACT with T cells engineered to express chimeric antigen receptors (CARs) has proven to be a highly effective strategy for the treatment of CD19-positive and BCMA-positive malignancies[1–3]. First-generation CARs aimed to mimic T cell activation by linking the intracellular signaling domain of CD3ζ to a single chain antibody (scFv)[4]. Next generation CARs have included one or more costimulatory molecules, such as CD28 or 4-1BB, upstream of CD3ζ[4,5]. These signaling components appear to successfully recapitulate signals 1 and 2 of T cell activation, although it is unclear whether these signals are subject to the same regulation as the native T cell receptor (TCR) and costimulatory receptors[6].

Synonymous with the clinical success of CAR-T cells in hematological malignancies[1,7–9] have been serious, and potentially lethal, toxicities including cytokine release syndrome, macrophage activation syndrome, hemophagocytic lymphohistiocytosis, and neurotoxicity[10–12]. Toxicities related to CAR-T cells are complex, multi-factorial, and manifest in a variety of ways[13–15]. Management of these toxicities has been a major concern for clinical implementation[12]. In contrast, ACT with T cell products (e.g., tumor-infiltrating lymphocytes (TIL) or TCR-engineered T cells) that rely on TCR signaling have reported low rates of adverse events relative to CAR-T cells[16]. Thus, the serious toxicities observed in the CD19 CAR-T cell clinical trials may be a specific feature of second-generation CAR-T cells, rather than T cell therapies in general.

We hypothesized that CAR toxicity is linked to the synthetic nature of the receptor design. As a strategy to redirect T cells in a TCR-dependent, MHC-independent manner, we created an alternative receptor, the T cell antigen coupler (TAC), which has three components: (1) an antigen-binding domain, (2) a TCR-recruitment domain, and (3) a co-receptor domain (hinge, transmembrane, and cytosolic regions). Since TAC receptors operate through the native TCR, we hypothesized they would induce a more controlled T cell response.

Here, we describe the modular design and functional characterization of TAC receptors. We present experimental evidence for the compatibility of the TAC platform with different classes of functional domains. Furthermore, we demonstrate the efficacy and unique biology of TAC-engineered human T cells in pre-clinical models of solid and hematological tumors. Notably, using a solid tumor model, we observe that TAC-engineered T cells display both enhanced in vivo anti-tumor efficacy and decreased off-tumor toxicity compared to first- and second-generation CARs.

## Results

**Selection of the TCR recruitment domain.** The TAC receptor was designed to trigger aggregation of the native TCR following binding of tumor antigens by co-opting the native TCR via the CD3 binding domain (Fig. 1). To evaluate the influence of CD3 binding on TAC receptor function, multiple anti-CD3 single-chain antibodies (scFvs) were evaluated, including UCHT1[17], huUCHT1[18,19], OKT3[20], L2K[21], and F6A[22]. These scFvs, which differ in their recognition of the ε chain[17,22–24], were assessed in the context of a TAC containing the CD4 co-receptor domain and various tumor-targeting moieties (Fig. 2a, e).

OKT3 and UCHT1 were evaluated in a HER2-specific TAC using a designed ankyrin repeat protein (H10-2-G3 DARPin[25]) as an antigen-binding domain. TACs employing OKT3 and UCHT1 displayed comparable levels of surface expression (Fig. 2b). Despite high surface expression, stimulation via the OKT3-TAC elicited low level cytokine production and poor cytotoxicity

(Fig. 2c, d). In contrast, antigen stimulation via the UCHT1-TAC triggered robust cytokine production and cytotoxicity (Fig. 2c, d). Using a TAC directed against CD19 via the FMC63 scFv[26], we also evaluated a humanized version of UCHT1 (huUCHT1)[18,19] and two other scFvs, L2K[21] and F6A[22] (Fig. 2e). The huUCHT1-TAC displayed the highest surface expression (Fig. 2f). The F6A-TAC was not detected on the T cell surface (Fig. 2f), despite successful detection on the surface of 293T cells (Supplementary Fig. 1A). Following stimulation with CD19-positive targets, we observed that significantly higher frequencies of CD4+ T cells engineered with huUCHT1-TAC produced cytokines (IFN-γ, TNF-α, IL-2), compared to CD4+ T cells engineered with F6A- or L2K-TACs (Fig. 2g). CD8+ T cells engineered with huUCHT1- and F6A-TAC produced comparable amounts of cytokine whereas stimulation via L2K-TACs elicited low levels of cytokine production (Fig. 2g). T cells engineered with the huUCHT1-TAC displayed markedly increased cytotoxicity relative to T cells engineered with either the F6A- and L2K-TAC (Fig. 2h). The ability of F6A-TAC to trigger cytokine production and cytotoxicity in an antigen-specific manner suggests that it was expressed on the T cell surface (Supplementary Fig. 1B) but was below the limit of detection. Compared to L2K, OKT3, and F6A, UCHT1 demonstrated preferred properties in the TAC platform and was employed in all subsequent TAC designs.

The CD3-binding domain was absolutely required for TAC activation (Supplementary Fig. 2A–D). Importantly, in the absence of antigen, we did not observe auto-activation of TAC-T cells (Supplementary Fig. 3A–C). To determine whether the CD8 co-receptor could be swapped in to replace the CD4 transmembrane and cytosolic domain, a TAC variant was generated using the CD8α co-receptor. To limit dimerization potential, we mutated two CD8α cysteine residues (C164S/C181S) creating the CD8-TAC. Direct comparisons of TAC receptors employing UCHT1 and either the CD4 or CD8 co-receptor revealed no functional differences (Supplementary Fig. 4A–D).

**TAC-T cells display no signs of auto-activation.** Auto-activation has been noted in the CAR literature causing expression of checkpoint receptors in the absence of antigenic stimulation[27,28]. T cells were engineered with a TAC receptor or a second-generation CAR containing either a CD28 (28ζ) or a 4-1BB (BBζ) costimulatory domain. All constructs utilized the same anti-HER2 DARPin targeting element. Consistent with previous reports[27], both CD28 and 4-1BB-based CAR-T cells showed significant elevation of checkpoint receptor expression relative to TAC-T cells (Fig. 3a). We observed upregulation of PD-1, LAG-3 and TIM-3 on CD28ζ CAR-T cells. T cells engineered with the BBζ CAR revealed significant elevation of LAG-3 but reduced expression of PD-1 and TIM-3 compared to the CD28-based CAR. In all cases, we observed donor-dependent effects where certain donors were more susceptible to checkpoint receptor upregulation (Supplementary Fig. 5A). We did not observe significant elevation of checkpoint receptors on HER2-TAC-T cells (Fig. 3a, Supplementary Fig. 5A). Since tonic signaling would also be expected to affect T cell differentiation, we performed a phenotypic analysis using the following markers: CD45RA, CCR7, CD28, and CD27. HER2TAC-T cells maintained a phenotypic profile most akin to control T cells (tNGFR), retaining similar proportions of naïve (CD45RA+, CCR7+), central memory (CD45RA−, CCR7+), effector memory (CD45RA−, CCR7−), and terminal effector T cells (CD45RA+, CCR7−) following the culture period (Fig. 3b). CD27 and CD28 costimulatory receptors were markedly downregulated on both 4-1BB- and CD28-based CAR-T cells, providing additional evidence of tonic signaling by these CARs (Fig. 3c). Interestingly, CD27 downregulation was

more pronounced on T cells engineered with the BBζ CAR than the 28ζ CAR while CD28 downregulation was more pronounced on T cells engineered with the 28ζ CAR, indicating a divergence in the tonic signals delivered by each intracellular domain. Consistent with our functional data (Supplementary Fig. 3), phenotypic profiling of checkpoint receptors and memory markers revealed no evidence of auto-activation by TAC-T cells.

**TAC-T cells show efficacy in solid and liquid tumor models.** We first evaluated the efficacy of TAC-T cells against OVCAR-3, a solid tumor xenograft that expresses HER2[29] (Fig. 4a). Tumor-bearing mice treated with HER2-TAC-T cells experienced rapid tumor regressions within 4–5 days following ACT, compared to mice treated with either CD19-TAC-T cells or T cells engineered with vector only (Fig. 4b). In all cases ($n = 11$), long-term tumor

control was observed, with mice surviving to the experimental endpoint (~60 days post-ACT).

TAC-T cells were tested against NALM-6 tumors, an acute lymphoblastic leukemia xenograft that expresses CD19[30] (Fig. 4c). Mice were treated with CD19-TAC-T cells or T cells engineered with a TAC lacking an antigen-binding domain (Fig. 4d). NALM-6 tumors grew rapidly in the mice treated with the TAC that lacked a binding domain causing significant morbidity within 3 weeks following treatment. Treatment with CD19-TAC-T cells caused rapid regression of NALM-6 tumors and the tumors were kept at a minimal level throughout the 60-day evaluation period in a majority of the mice treated (Fig. 4d).

**TAC-T cells show enhanced activity in a solid tumor model.** We compared HER2-TAC-T cells to T cells engineered with

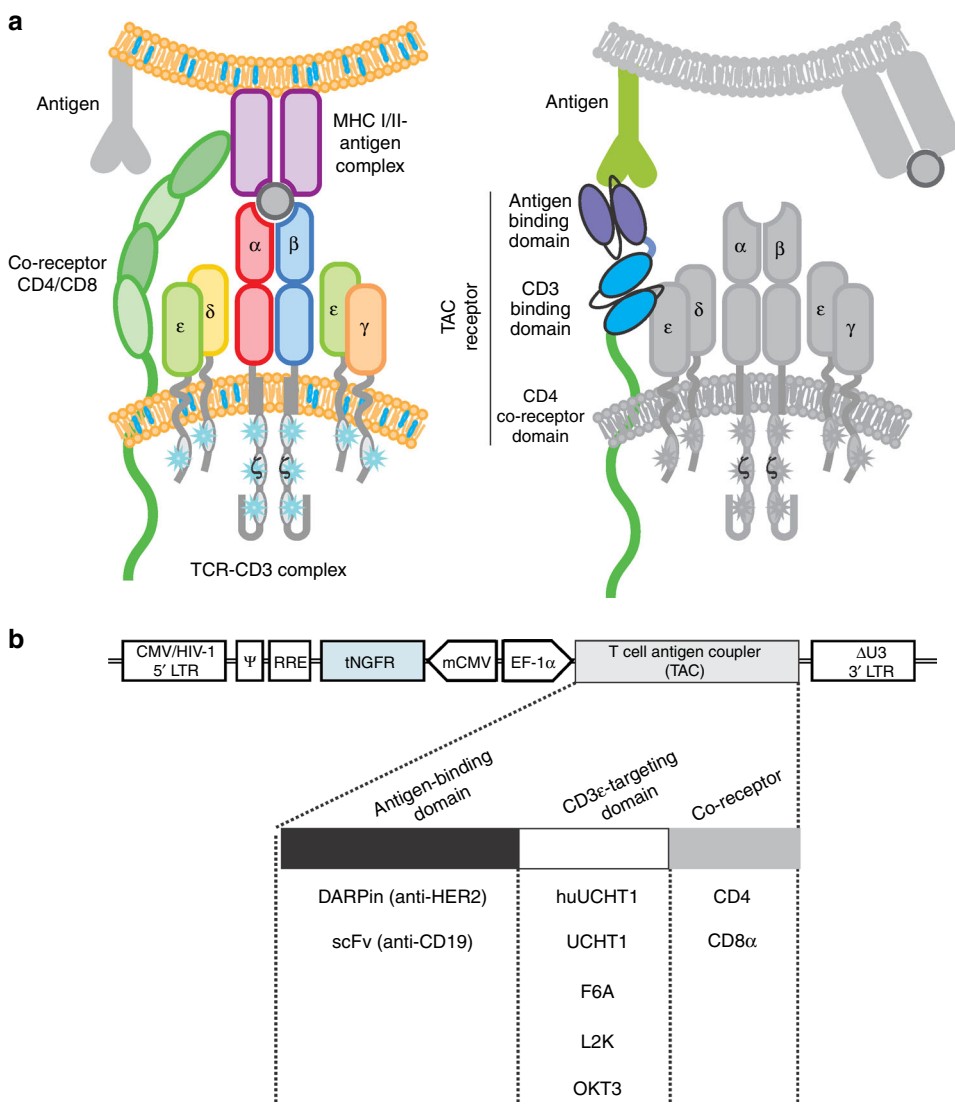

**Fig. 1** TAC design mimics the TCR-CD3:co-receptor complex. **a** Left: Naturally occurring TCR-CD3 complex interacts directly with the antigen presented by MHC. Meanwhile, the CD8/CD4 co-receptor interacts with MHC I/II in an antigen-independent manner. Together, these interactions comprise the first step in T cell activation. Right: The TAC receptor re-directs the TCR-CD3 complex towards an antigen of choice using an interchangeable antigen binding moiety (here depicted with an scFv, purple). An scFv is used to recruit the TCR-CD3 complex (blue). Co-receptor properties are incorporated by including the CD4 hinge, TM region, and cytosolic tail (green). **b** The TAC is incorporated into the pCCL DNA backbone containing a truncated NGFR (tNGFR), which lacks cytosolic signaling domains, as a transduction control. The vector features a bi-directional promoter system with tNGFR under control of the mCMV promoter and TAC expression being driven by the EF-1α promoter. TAC is comprised of an antigen binding domain, a CD3-binding domain, and a co-receptor domain. A variety of proteins can be used for each of these three TAC domains allowing the TAC to be modified to best respond to numerous different antigens. The specific domain combinations tested are described below

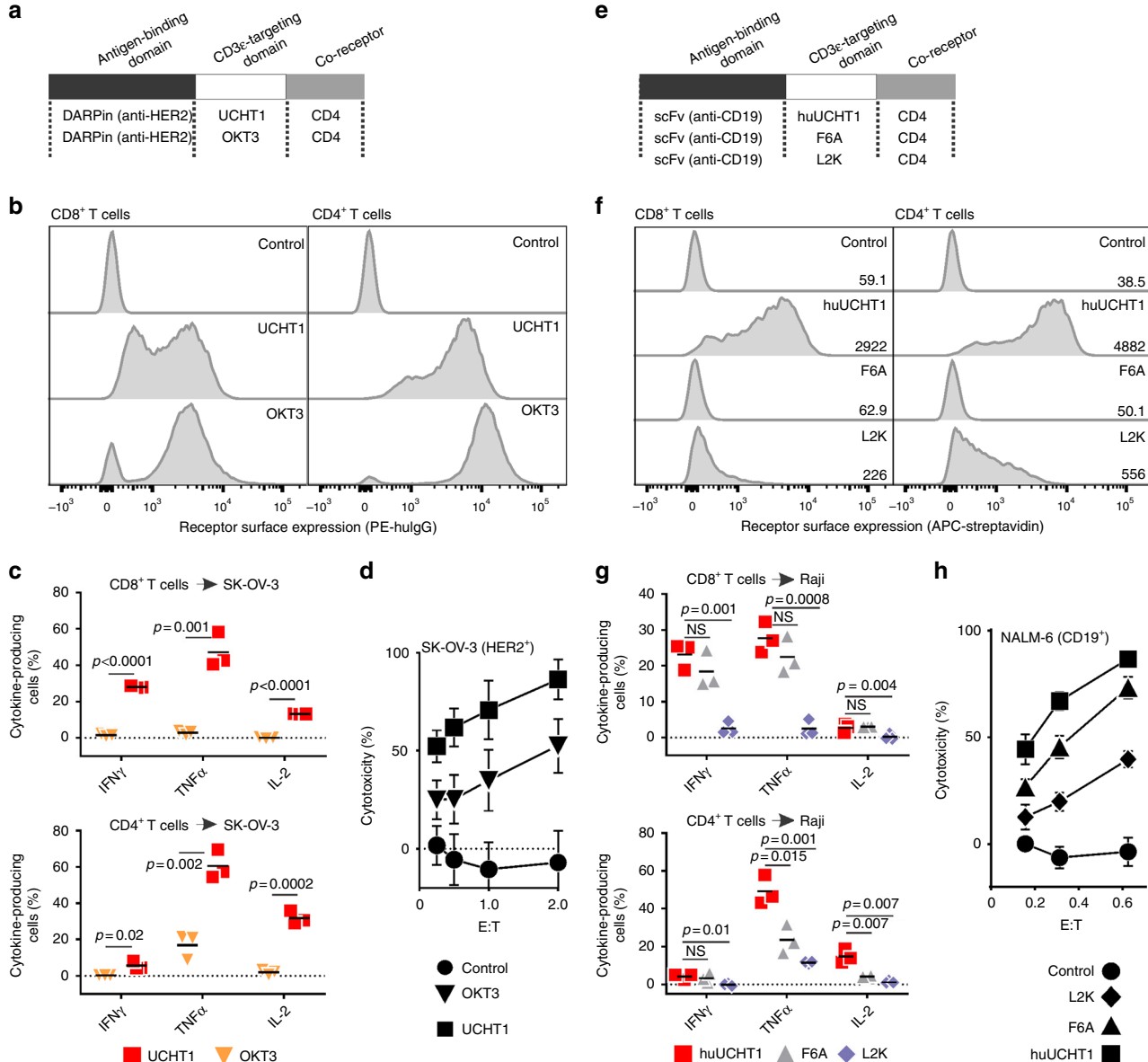

**Fig. 2** Evaluation of multiple anti-CD3 scFv domains for recruitment of TAC to the TCR-CD3 complex. **a**, **e** Schematic representation of evaluated TAC constructs. TAC receptors utilizing the (**a**) anti-HER2 DARPin are paired with either the UCHT1 or OKT3 anti-CD3 scFv. TAC receptors using the (**e**) anti-CD19 scFv are paired with either the huUCHT1, F6A, or L2K anti-CD3 scFv. **b**, **f** Relative TAC surface expression is measured by flow cytometry. Cells are stained for CD4, CD8, tNGFR and TAC, and gated on either CD4⁺NGFR⁺ or CD8⁺NGFR⁺. Representative data of three independent experiments are presented as histogram analysis of (**b**) HER2-TAC or (**f**) CD19-TAC. Surface expression of OKT3 relative to UCHT1 was significantly higher in CD4 cells ($p = 0.0007$) but not in CD8 cells. huUCHT1 expression is significantly higher compared to either L2K ($p = 0.005$ (CD4)/0.0002 (CD8)) or F6A ($p < 0.0001$ (CD4) $p < 0.0001$ (CD8)). For the gating strategy see Supplementary Fig. 13A. **c**, **g** HER2- and CD19-specific TAC-T cells are stimulated with antigen-positive (**c**) SK-OV-3 and (**g**) Raji tumor cells, respectively. Data are presented as percent of CD4 or CD8 T cells producing cytokine. Cytokine producing cells are compared from (**c**) TAC-T cells bearing UCHT1 (square) or OKT3 (inverted triangle), or (**g**) TAC-T cells bearing huUCHT1 (square), F6A (triangle), or L2K (diamond). Lines represent the mean. Multiple *t*-test is used to determine significance in all cases. For the gating strategy see Supplementary Fig. 13B. **d**, **h** HER2- and CD19-TAC and vector control (vector only carrying tNGFR) T cells are co-cultured with (**d**) SK-OV-3 and (**h**) NALM-6 tumor cells, respectively, to measure TAC-T cell-mediated cytotoxicity. Vector control T cells (circles) are compared against **d** HER2-specific TAC-T cells bearing UCHT1 (square) or OKT3 (triangle), or (**h**) CD19-specific TAC-T cells bearing huUCHT1 (square), F6A (triangle), or L2K (diamond). Data are from three independent experiments with three different donors, error bars are standard deviation

either a first-generation or a CD28-based, second-generation HER2-CAR[31]; all chimeric receptors were targeted against HER2 with the same DARPin. All engineered T cells displayed comparable in vitro cytotoxicity and cytokine production (Supplementary Fig. 6A–F). These in vitro similarities notwithstanding, we observed marked differences in the anti-tumor activity of HER2-TAC- and HER2-CAR-engineered T cells in vivo

(Fig. 5a–d). Consistent with the earlier data, tumors treated with HER2-TAC-T cells regressed within a few days following T cell infusion (Fig. 5a). In striking contrast, first-generation HER2-CAR-T cells showed no anti-tumor activity and tumor progression was not significantly different from control treated mice (Fig. 5b). Second-generation HER2-CAR-T cells displayed only moderate, delayed anti-tumor efficacy which manifested around

3 weeks post-ACT (Fig. 5c). Tumor growth, following treatment with HER2-TAC-T cells, was found to be significantly different compared to treatment with first- and second-generation CAR-T cells. In contrast, treatment with first-generation CAR-T cells did not lead to significantly different ($p = 0.5$) tumor growth relative to control mice. The change in body weight for mice

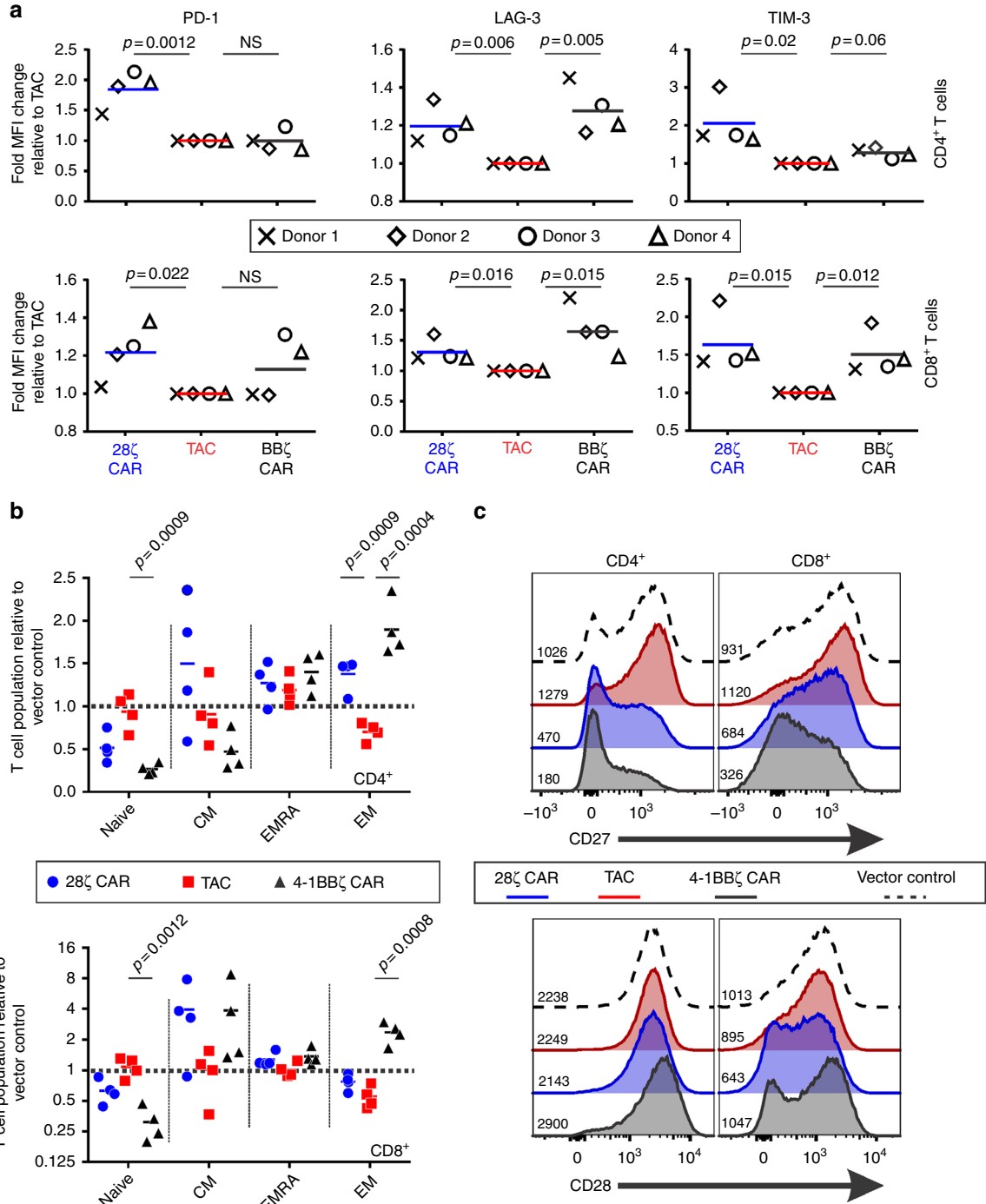

**Fig. 3** Relative expression of checkpoint receptors and memory T cell subsets in CAR- and TAC-engineered T cells. T cells were transduced with HER2-TAC, or a second-generation anti-HER2 CAR including the CD28 (28ζ CAR) or 4-1BB (BBζ CAR) costimulatory receptor domains, or a vector control (tNGFR). Engineered T cells are stained for surface marker expression and CD4$^+$NGFR$^+$ or CD8$^+$NGFR$^+$ populations are analyzed by flow cytometry for (**a**) expression of checkpoint receptors PD-1, LAG-3, and TIM-3. All data is normalized to TAC-engineered T cells, and lines represent the mean of four donors. Each donor is represented by a unique symbol to highlight donor-to-donor variations. Multiple $t$-test is used to determine significance. For the gating strategy see Supplementary Fig. 14. **b** Memory T cell subsets of TAC-, 28ζ CAR-, and BBζ CAR-T cells, relative to T cells engineered with a vector control (tNGFR). T cell subsets are defined as naïve (CD45RA$^+$, CCR7$^+$), central memory (CM) (CD45RA$^-$, CCR7$^+$), effector memory (EM) (CD45RA$^-$, CCR7$^-$), and terminal effectors (EMRA) (CD45RA$^+$, CCR7$^-$). Lines represent the mean of four donors. Multiple $t$-test is used to determine significance. **c** CD27 and CD28 expression; representative histograms show data from one donor, median fluorescence intensity is indicated. Data are from two independent experiments with four different donors. For both (**b**) and (**c**), the gating strategy is shown in Supplementary Fig. 15

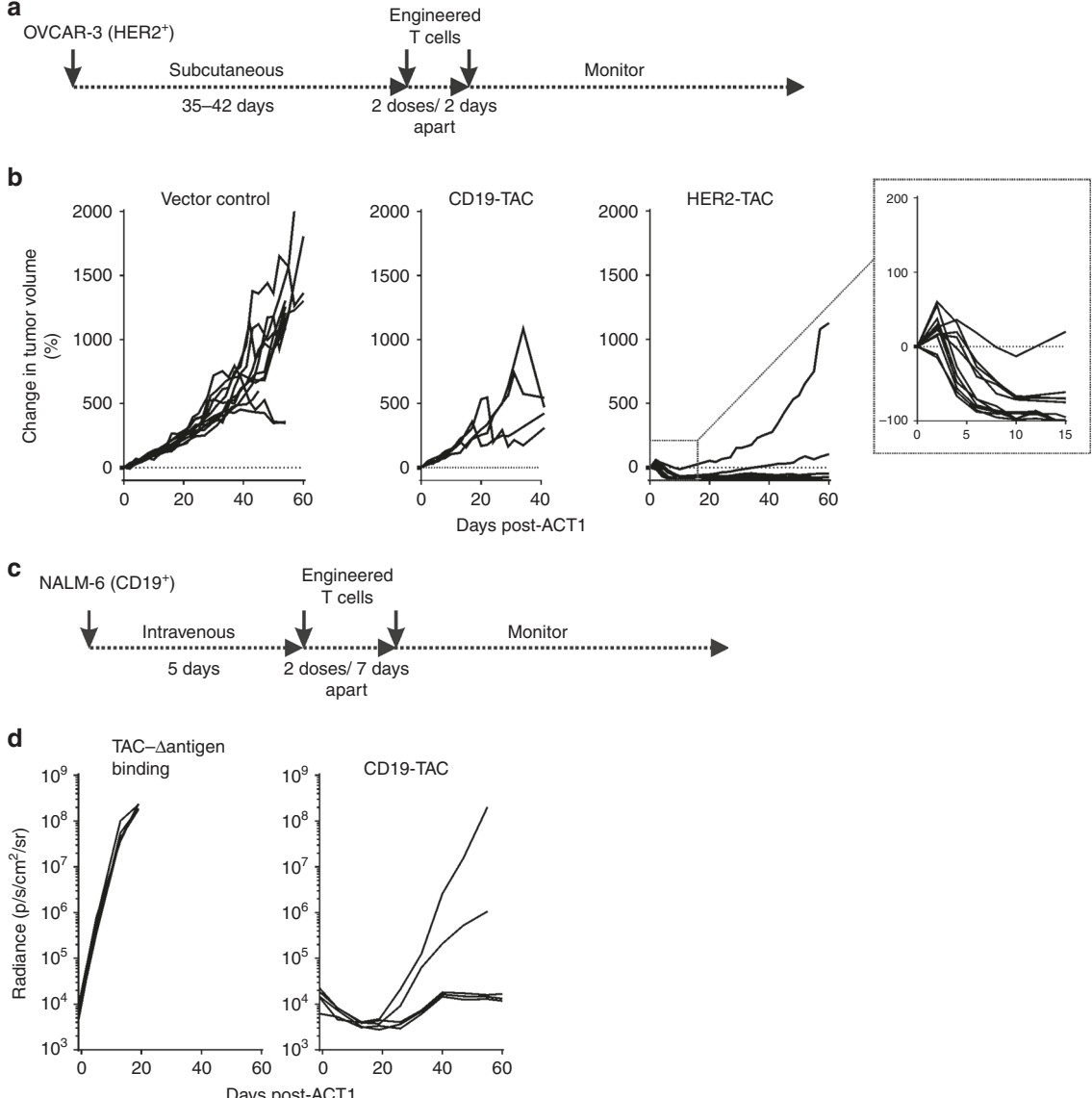

**Fig. 4** TAC-T cells demonstrate in vivo efficacy against solid and liquid tumors. **a** Treatment schema for OVCAR-3 tumor-bearing mice. In short, NRG mice receive $2.5 \times 10^6$ OVCAR-3 cells subcutaneously. Tumors grow for 35–42 days until an average size of ~100 mm³ is achieved. T cells are delivered over two doses, 48 h apart. **b** Tumor-bearing mice receive intravenous delivery of $4–6 \times 10^6$ HER2-TAC-T cells, CD19-TAC-T cells, or an equivalent dose of vector control T cells. Tumor growth is followed over time. Each curve shows data from a single treated tumor. Data are from three donors, collected over two independent experiments, $n = 11$ for each of HER2-TAC and vector control groups; CD19-TAC data generated from one donor, one experiment, $n = 4$. Using curve fitting analysis and multiple $t$-test HER2-TAC induced regression is significantly different from controls, while tumor growth between vector control and CD19-TAC was not significantly different. **c** Treatment schema for NALM-6 tumor-bearing mice. In short, 0.5 million NALM-6 cells are administered intravenously and allowed to establish for 5 days. Mice were treated with a total dose of $4 \times 10^6$ cryopreserved CD19-TAC-T cells. T cells are delivered over two doses, 7 days apart. **d** Mice are treated with either TAC-Δantigen-binding domain or CD19-TAC-T cells. Curves each represent a single treated tumor. Data are from one donor, one experiment, $n = 5$ for each of CD19-TAC and control groups. Data have been replicated in independent experiments, $n = 10$. Tumor progression is followed weekly via luminescence

treated with second-generation CAR was significantly different from the change in weight of mice treated with TAC-T cells, first-generation CAR-T cells, and control mice. The body weights of mice treated with vector control, first-generation CAR, or TAC were not significantly different (multiple $t$-test shows no significance; First-generation/TAC $p = 0.5$; First-generation/ control $p = 0.9$; TAC/Control $p = 0.5$).

In addition to efficacy, we compared the toxicities associated with TAC- versus CAR-engineered T cells. HER2-CAR-T cells elicited a profound toxicity that manifested as a ruffled coat,

labored breathing and decreased body condition. We employed weight loss (Fig. 5a–d) and core body temperature (Supplementary Fig. 7A–D) as a quantitative measure of toxicity. Mice treated with first-generation HER2-CAR-T cells did not exhibit signs of toxicity, consistent with the lack of anti-tumor efficacy (Fig. 5b). In contrast, we noted toxicities emerging following treatment with second-generation HER2-CAR-T cells were quite severe and all mice became moribund within 44 days of ACT (Fig. 5c, X marks the endpoint) (Supplementary Fig. 7E). Conversely, HER2-TAC-T cells clearly distinguish themselves from HER2-CAR-T

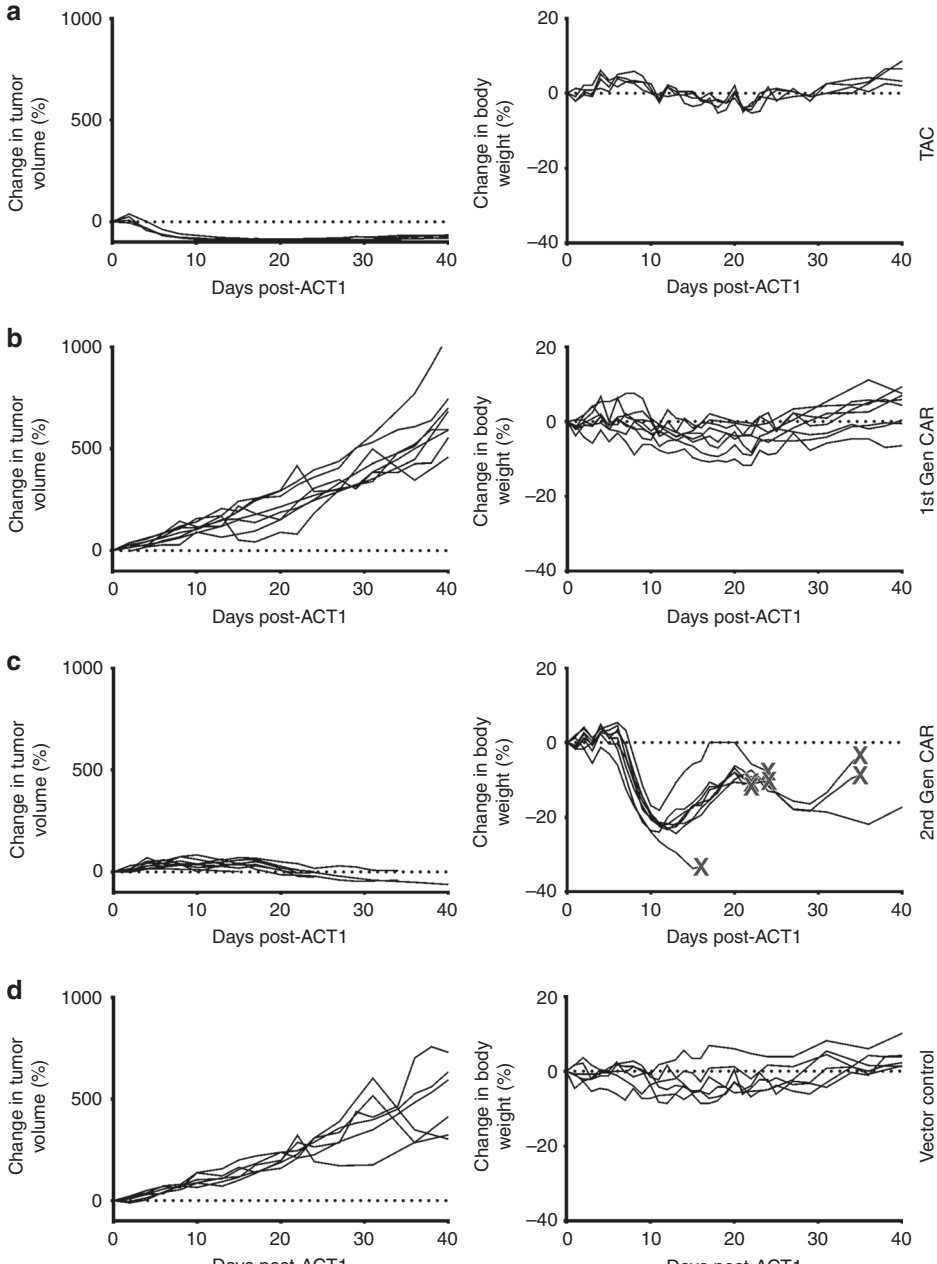

**Fig. 5** HER2-TAC-T cells demonstrate an enhanced safety profile and improved efficacy over first- and second-generation HER2-CAR-T cells in vivo. OVCAR-3 tumor-bearing mice are treated with 2.0 × 10⁶ HER2-TAC-T cells (**a**), first-generation anti-HER2 CAR-T cells (**b**), second-generation anti-HER2 28ζ CAR-T cells (**c**), or a matched total number of vector control T cells (**d**). Mice are followed for change in body weight and tumor volume; each curve represents a single treated mouse relative to pre-treatment weight/volume. When mice reach endpoint, this is indicated via X in (**c**). Data has been replicated in an independent experiment. Using curve fitting analysis and multiple *t*-test HER2-TAC induced regression is significantly different from control, 1st, and 2nd generation CAR curves, while tumor growth between vector control and 1st gen CAR was not significantly different. The change in body weight observed in 2nd gen CAR curves is significantly different from vector control, 1st gen CAR, and TAC-T cell treated mice. However, the difference in body weight between vector control, 1st gen CAR, and TAC curves was not significantly different

cells as the HER2-TAC-T cells did not trigger any observable toxicities while demonstrating robust anti-tumor efficacy (Fig. 5a and Supplementary Fig. 7A).

**TAC-T cells show greater tumor penetration than CAR-T cells**. To elucidate the differential effects of HER2-TAC- and CAR-T cells in vivo, mice treated with equal doses of HER2-TAC- or second-generation CD28-based HER2-CAR-T cells were subjected to pathologic and serum cytokine analysis at 1, 3, 5,

and 7 days post-adoptive transfer (Fig. 6 and Supplementary Figs. 8–10). Mice receiving CD28-based HER2-CAR-T cells developed severe pulmonary pathology. Masses of infiltrating leukocytes formed at perivascular (Supplementary Fig. 8A) and sub-pleural sites, becoming progressively larger over time. Immunohistochemistry (IHC) showed that the pulmonary deposits were largely composed of CD3+ T cells (Supplementary Fig. 8B). A patchy leukocytic infiltrate containing CD3⁺ cells was also present in the cardiac tissue (Supplementary Fig. 8C). In contrast, HER2-TAC- or control T cells showed only scattered

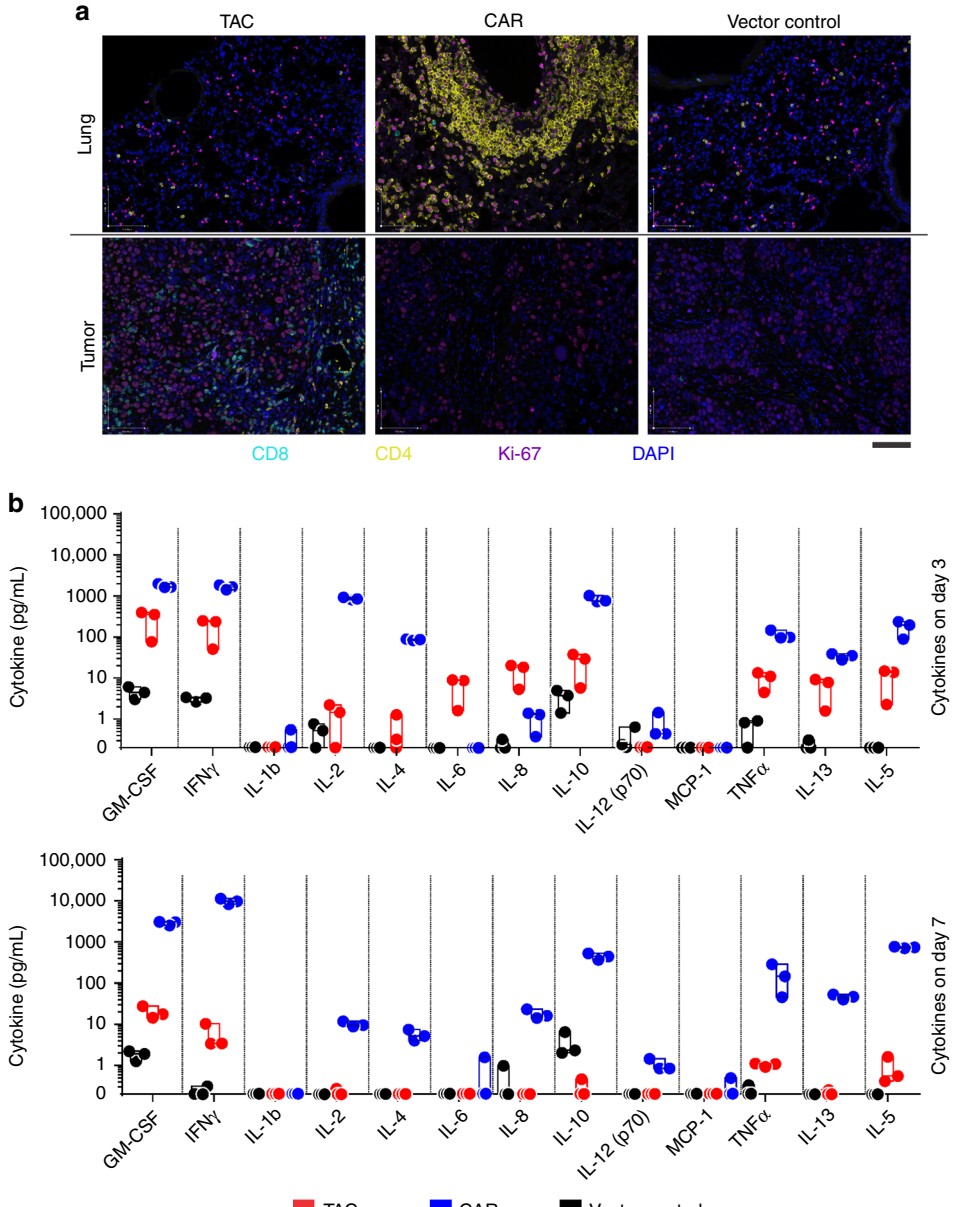

**Fig. 6** Engineered T cell distribution and cytokine release in vivo. OVCAR-3 tumor-bearing mice were treated with 6.0 × 10⁶ anti-HER2 28ζ CAR- or HER2-TAC-T cells, or a matched total number of vector control cells. Mice are sacrificed at 1, 3, 5, and 7 days post-ACT1 for multiplex serum cytokine analysis or perfusion and fixation of tissues for subsequent histology. **a** Multicolor IHC is performed on tumor and lung tissue 7 days post-ACT1. Tissues are stained for CD8 (cyan), CD4 (yellow), DNA (blue) and a proliferation marker (Ki-67, purple). Data are representative of 3 mice. Scale bar indicates 100 μm. **b** Multiplex analysis of human cytokines in mouse serum on day 3 and 7. Measurements that fall below 0.2 pg/mL are below the calibration range and are therefore defined as 0. 0 values are depicted on the graph's x axis. Statistical analysis is provided in Supplementary Table 1, analysis was performed using multiple *t*-test. Individual data points are shown, bars indicate standard deviation and center bars indicate the median

pulmonary and cardiac infiltrates (Supplementary Figs. 8A, B and 10). Instead, HER2-TAC-T cells accumulated primarily within tumor tissue, whereas little accumulation of HER2-CAR-T cells or control T cells was observed within the experimental time frame (Supplementary Fig. 8D, E), consistent with the delayed tumor control exhibited by the HER2-CAR-T cells.

To further define the nature of the T cell infiltrate within tissues, we employed multiparametric IHC. Lung sections from mice treated with HER2-CAR-T cells revealed a dominant infiltration of Ki-67 + CD4+ T cells, suggesting localized expansion (Fig. 6a, Supplementary Fig. 8F). In contrast, the few CD4+ and CD8+ T cells observed in the lungs of mice treated with HER2-TAC-T cells or vector control T cells were quiescent

based on their small size (Supplementary Fig. 8B) and relative lack of Ki-67 staining (Fig. 6a). Similar observations were made in cardiac tissue (Supplementary Figs. 8C, 10). In tumors, the HER2-TAC-T cell population was dominated by CD8+ T cells, although CD4+ T cells were also evident. Further, intra-tumoral HER2-TAC-T cells were Ki-67+, indicating localized proliferation (Fig. 6a). Tumor tissue infiltrated by HER2-TAC-T cells also contained necrotic tumor cells indicative of cytotoxicity (Supplementary Fig. 8E). Findings regarding tissue-specific CD3+ T cell infiltration were quantified by blinded scoring of IHC slides (Supplementary Fig. 8F).

Examination of circulating human cytokines in the serum of treated mice demonstrated a striking discordance in the quantity

and chronology of inflammatory cytokine production between HER2-TAC- and HER2-CAR-T cells. HER2-CAR-T cell-treated mice exhibited the highest levels of circulating cytokines when compared to receptor-negative or TAC-T cell-treated mice (Fig. 6b). Moreover, while serum cytokine levels in HER2-TAC-T cell-treated mice remained low throughout treatment, serum cytokine levels in HER2-CAR-T cell-treated mice became progressively higher over the course of treatment (Fig. 6b).

In other experiments, we determined that the HER2-CAR-T cells also cause toxicity in tumor-free animals, indicating the toxicity is due, at least in part, to antigen expression in healthy tissues. No cross reactivity with murine HER2 was observed (Supplementary Fig. 11). Thus, cells were responding against an unknown antigen in the lungs and heart driving a lethal systemic inflammatory reaction. In contrast, HER2-TAC-T cells, which carry the same antigen-recognition domain, showed improved penetration and expansion within the tumor and did not show evidence of activation or pathology within the lung and heart.

## Discussion

Novel strategies, such as splitting of activation and costimulation signals[32], and Boolean-gated receptors[33], seek to improve the toxicity profile of CAR-T cells. Rather than modifying the CAR design, we opted to design a major histocompatibility complex (MHC)-independent receptor that recapitulates the architecture of a TCR-CD3/co-receptor complex to engage natural cellular pathways and achieve a more nuanced T cell response.

Our data show that engagement of the TCR-CD3 complex is crucial for the function of TAC-engineered T cells. The activity of the TAC receptor was critically dependent upon the choice of CD3-binding domain, with UCHT1 demonstrating the strongest combination of phenotypic and functional characteristics in vitro. Curiously, the scFv derived from OKT3, one of the most commonly used agonist anti-CD3 antibodies, performed poorly in the TAC platform. Despite the overlapping epitopes of OKT3 and UCHT1[17], our findings, and those of previous studies[23], indicate that small differences in binding to the CD3 complex can result in substantially different functional outcomes.

Regardless of whether the anti-CD3 scFv binds a complex structural epitope (UCHT1[34–36], OKT3[23,34,37], and L2K[21]) or a simple amino acid sequence (F6A binds to a linear N-terminal CD3ε epitope, AA 22–26 "QDGNE"[22]), all CD3-recruiting scFvs displayed some level of functionality within the TAC framework. The varying functionalities of the TAC receptors carrying different CD3-binding moieties suggest that these variabilities could be used to fine-tune T cell responses by altering the various modules of the TAC to create an appropriate indication-specific receptor.

Since the use of murine-derived scFvs in chimeric receptors has been associated with the generation of human anti-mouse antibodies that eliminate engineered T cells[38,39], we have validated the use of humanized UCHT1 in the TAC receptor and all future iterations of the TAC will employ the human scaffold. Our proof-of-concept studies have focused on TAC receptors directed at either CD19 or HER2, but in principle, any cell surface target should be amenable to TAC recognition.

Our in vitro comparison of TAC- and first- and second-generation CAR-T cells revealed no functional differences in cytotoxicity or cytokine production. We noted that both CD28- and 4-1BB-based second-generation CARs, but not TAC receptors, delivered tonic signals to T cells, which manifested in elevated expression of checkpoint receptors and reduced frequencies of T cells with a naive phenotype in the manufactured product. Given recent reports that indicate tonic signaling can impair CAR-T cell function[27,28], the lack of tonic signaling may be an advantage to TAC receptors. Tonic signaling in CARs can be exacerbated by the choice of scFv[27,40]. We have employed multiple scFvs with the TAC platform and failed to observe evidence of tonic signaling (Supplementary Fig. 12). We believe that the lack of tonic signaling may be due to the lack of immunoreceptor tyrosine-based activation motifs (ITAMs) in the TAC receptor. It has been suggested that less differentiated T cells are preferable to terminally-differentiated effector cells for adoptive therapy, as they retain greater proliferative ability and improved in vivo persistence[41]. Data reveal that T cells engineered with TAC receptors retain a less differentiated phenotype, which may translate to a more potent T cell product.

Based on the 2-signal hypothesis of T cell activation[42], one could expect TAC-T cells to perform similarly to a first-generation CAR. However, TAC-T cells showed superior outcomes in vivo compared to both first- and second-generation CARs. Pathological analysis revealed two important features of TAC-T cells: (1) greater infiltration of solid tumors post-adoptive transfer and (2) reduced expansion in healthy tissues that express antigens that trigger the HER2-CAR. Intratumorally, both CD4+ and CD8+ TAC-T cells expanded, demonstrating a balanced anti-tumor attack. Importantly, TAC-T cells did not show evidence of activation or expansion within the lungs, heart or any other tissue and did not cause any other toxicities.

Looking at CAR-T cell mediated toxicities, the incidence and severity of clinical adverse events vary widely across CAR-T cell trials. In some trials, 100% of treated patients experienced toxicities, including fever, nausea, general malaise, and in rare cases lethality[43,44]. The solid tumor model we employed enables simultaneous monitoring of CAR-T cell-mediated efficacy and toxicity. Therefore, it is intriguing that, in addition to superior solid tumor control, TAC-T cells also displayed less toxicity than CAR-T cells. In contrast, HER2-CAR-T cells infiltrated normal lung and heart tissues, resulting in robust expansion of CD4+ CAR-T cells. Examination of serum cytokines following infusion of the second-generation CAR-T cells revealed exuberant production of a range of cytokines, indicating the expansion was not reflective of a specific CD4+ T cell subset[45]. In contrast, circulating cytokines following infusion of TAC-T cells were markedly lower and biased towards a Th1 cytokine profile, suggesting a more controlled response.

It is curious that TAC-, but not CAR-, T cells infiltrated the tumor tissue at early time points post-ACT and, conversely, that CAR-, but not TAC-, T cells expanded greatly within the lungs and heart. The lungs, as the first-pass organ, could be expected to be more susceptible to off-tumor reactivity of engineered T cells; however, CAR-T cell expansion within the heart argues that the infiltration/pathology is not simply a first-pass effect. We observed some CD8+ and CD4+ T cells in the lung following infusion of TAC-T cells. However, their small size and lack of Ki-67 expression indicated that they were quiescent. It remains to be determined why TAC-T cells did not react to these healthy tissues as exuberantly as the CAR-T cells. Regardless, these results demonstrate that while TACs can engage and eliminate antigen-bearing tumor cells, they are also sufficiently selective to bypass healthy cells bearing low levels of antigen. This ability to differentiate between antigen in healthy and cancerous tissues could, if generalizable, allow TAC-T cells to be used with solid tumor antigens that are expressed at low levels on healthy cells.

Our observations with HER2-TAC-T cells demonstrate that the efficacy of engineered T cells can be uncoupled from toxicity. Importantly, we predict that this profile will significantly reduce risk and improve tolerability in patients—including those with significant comorbidities. Furthermore, the improved safety profile would make TAC therapy accessible to a much larger pool of patients, as it would no longer be limited to academic centers

capable of handling the complex toxicities currently encountered with CAR therapies. For these reasons, the use of TAC in clinical applications is highly anticipated.

## Materials and methods

**CAR and TAC vector generation.** TAC receptor transgenes were designed by linking tumor-directing moiety to a CD3-TCR complex-targeting single chain variable fragment (scFv), and the hinge, transmembrane (TM), and cytoplasmic domains of a T cell co-receptor.

The TAC sequence using UCHT1, CD4 hinge, transmembrane, and cytoplasmic domains was synthesized from GeneArt (Invitrogen; Thermo Fisher Scientific, Waltham, MA) in the pUC57 vector. The HER2-specific H10-2-G3 DARPin[31] targeting domain (using an Igκ leader sequence) was PCR amplified (Primers FWD-1 and REV-1 Supplementary Table 2) and cloned into the pUC57 TAC vector using AscI and XmaI cut sites. The resulting TAC was then cloned into the pCCL vector, containing bi-directional minimal CMV (mCMV) and EF-1α promoters[46] (kindly provided by Megan Levings, University of British Columbia, Vancouver, BC), using AscI and NheI cut sites.

To generate the OKT3 TAC the $V_L$-$V_H$ configuration of the OKT3 Q/S variant[20] was ordered from GeneArt and cloned into pUC57 using BamHI and SpeI. The resulting TAC construct was then sub-cloned into pCCL as above.

TAC -ΔAntigen binding site was cloned from the HER2 DARPin TAC construct where the HER2 binding domain was removed via overlapping PCR using primers FWD-5/6 and REV-5/6 (Supplementary Table 2).

Generation of the second-generation anti-HER2 28ζ CAR consisting of the Igκ leader, anti-HER2 H10-2-G3 DARPin, human myc tag, CD8α hinge, CD28 TM and cytoplasmic domains, and CD3ζ cytoplasmic tail was previously described[1]. The second-generation anti-HER2 BBζ CAR (consisting of the Igκ leader, anti-HER2 H10-2-G3 DARPin, human myc tag, BamHI site, CD8α hinge and TM, 4-1BB cytoplasmic domain, CD3ζ cytoplasmic tail, and NheI site) was generated by cloning the CD8α hinge and TM, 4-1BB cytoplasmic domain, and CD3ζ cytoplasmic tail portions from an anti-CD19 CAR (prepared according to[47]) between the BamHI and NheI sites of the anti-HER2 28ζ CAR. (Primers FWD-4 and REV-4 Supplementary Table 2). The first-generation anti-HER2 CAR was constructed by removing the 4-1BB portion of the anti-HER2 BBζ CAR. This was completed by deletion PCR (Primers FWD-2 and 3 and REV-2 and 3; Supplementary Table 2).

For anti-BCMA CAR and TAC receptors, the anti-BCMA scFv (C11D5.3) sequence was obtained from patent US20150051266A1[3] and synthesized by Genscript. The scFv was sub-cloned into existing pCCL TAC and CD28ζ-CAR lentiviral backbones using AscI and BamHI cut sites, replacing the existing scFv with C11D5.3.

The TAC encoding the humanized version of UCHT1 (huUCHT1)[18,19] was ordered from GeneArt and sub-cloned by the manufacturer into the pCCL TAC backbone we provided. The sequence for the scFv derived from FMC63, a CD19-specific monoclonal antibody[26] was synthesized with a N-terminal CD8α leader sequence and a Whitlow linker sequence between the heavy and light variable fragments[48] (Integrated DNA Technologies Inc., Coralville, AI). The FMC63 scFv was amplified via the TOPO cloning kit (Invitrogen), then sub-cloned into the huUCHT1 TAC pCCL backbone using AscI and BamHI. TAC constructs using the F6A[22] or L2K[21] CD3-binding domain in place of the huUCHT1 were ordered from GeneArt.

The CD8α TAC was cloned as follows: the CD8α sequence was obtained from the UniProtKB/Swiss-Prot database (entry: P01732) and ordered from GeneArt containing C164S and C181S mutations within the hinge domain. The sequence was cloned into the pUC57 UCHT1 TAC vector using XhoI and NheI. The resulting CD8α-TAC was then sub-cloned into the pCCL vector using AscI and NheI.

All restriction enzymes were purchased from New England BioLabs (NEB, Whitby, ON). All sequences were codon-optimized for expression in human cells and verified. All TAC and CAR constructs are under the control of the human EF-1α promoter through 5′ AscI and 3′ NheI cut sites. Truncated LNGFR (tNGFR) under control of a minimal human cytomegalovirus (mCMV) promoter was utilized as a transduction marker. The receptor-negative control vector codes only for the tNGFR transgene under the mCMV promoter.

**Lentivirus production.** Self-inactivating, non-replicative lentivirus produced using a third-generation system has been previously discussed[46,49]. Briefly, $8 \times 10^6$ HEK293T cells cultured on 15 cm diameter tissue culture-treated dishes (NUNC; Thermo Fisher Scientific) were transfected with the packaging plasmids pRSV-Rev (6.25 μg), pMD2.G (9 μg), pMDLg-pRRE (12.5 μg) and the transfer plasmid pCCL containing the transgene (32 μg) using Opti-MEM (Gibco; Thermo Fisher Scientific) and Lipofectamine 2000 (Thermo Fisher Scientific). Twelve to sixteen hours after transfection, media was replenished with new medium supplemented with sodium butyrate (1 mM; Sigma-Aldrich Canada Co., Oakville, ON). Media containing lentivirus particles was collected after 36–48 h and concentrated by ultra-centrifugation. Viral titer in TU/mL was determined by serial dilution and transduction of HEK293T cells, and subsequently determining %tNGFR+ via flow cytometry using an anti-NGFR-VioBrightFITC antibody (Miltenyi Biotec, Bergisch

Gladbach, Germany). CD19-TAC was manufactured by Lentigen (Lentigen Corp, Gaithersburg, USA). The virus was shipped in frozen form, thawed, re-frozen and titered.

**Transduction of human T cells.** This research was approved by the McMaster Health Sciences Research Ethics Board and all donors in this study provided informed written consent. Receptor-engineered human T cells were generated as previously described[49]. Human peripheral blood mononuclear cells (PBMC) from healthy donors (McMaster Adult Cohort (MAC) donor) or commercial leukapheresis products (LEUK donor) (HemaCare, Van Nuys, CA) were isolated by Ficoll-Paque-Plus gradient centrifugation (GE Healthcare, Baie d'Urfe, QC) and cryopreserved in inactivated human AB serum (Corning, Corning, NY) containing 10% DMSO (Sigma-Aldrich Canada Co.).

Bulk T cells were activated from PBMCs with anti-CD3/28 Dynabeads at a 0.8:1 bead-to-cell ratio (Gibco) following manufacturer's guidelines, and were cultured in RPMI 1640 (Gibco) supplemented with 8.7% heat-inactivated fetal bovine serum (Gibco), 1.75 mM L-glutamine, 8.7 mM HEPES, 0.87 mM sodium pyruvate (Sigma-Aldrich Canada Co.), 0.87 × non-essential amino acids (Gibco), 48 μM β-mercaptoethanol, 87 U/mL penicillin + 87 μg/mL streptomycin, 660 I.U. rhIL-2 and 10 ng/mL rhIL-7 (PeproTech, Rocky Hill, NJ). After 18–24 h, cells were transduced with lentivirus at an MOI between 1–5 (CAR or tNGFR) or 10 (TAC). Cells were monitored daily and fed according to cell counts every 2–3 days for a period of 11–14 days prior to use in vitro and/or in vivo.

**Phenotypic analysis by flow cytometry.** Engineered T cell surface expression of CD4, CD8, and where applicable tNGFR was evaluated through direct staining with conjugated antibodies. All stains were carried at room temperature for 30 min. HER2-specific CAR- or TAC-T cells were first stained with rhHER2-Fc chimera protein (R&D Systems, Minneapolis, MN), followed by conjugated antibodies against CD4, CD8, NGFR (BD Biosciences, San Jose, CA; eBioscience, Thermo Fisher Scientific) and human IgG (Jackson ImmunoResearch, West Grove, PA). CD19-specific TAC-T cells were first stained with biotinylated Protein L (Thermo Fisher Scientific), followed by streptavidin-APC (BD Biosciences), and finally conjugated antibodies against CD4, CD8, and NGFR (BD Biosciences). All flow cytometry was conducted on a BD LSRFortessa or BD LSRII cytometer (BD Bioscience) and analyzed using FlowJo vX software (FlowJo, LLC, Ashland, OR). Expression of murine or human HER2 on tumor cell lines was determined by either direct staining with allophycocyanin (APC)-conjugated anti-mHER2 (clone: 666521, Cat No. FAB6744A, R&D Systems) or a two-step stain by indirect immunofluorescence (conducted at 4 °C); incubation with Herceptin (anti-hHER2; kind gift from Ronan Foley, McMaster University, Hamilton, ON) was followed by PE-goat-anti-human IgG (as above). Flow cytometry gating strategies are included in Supplementary Figs. 13–16.

**Functional analysis of CAR-T cells.** A total of $5 \times 10^5$ or $4 \times 10^5$ engineered T cells were stimulated with $5 \times 10^4$ antigen-positive or -negative tumor cells for 4 h at 37 °C in a round- or flat-bottom 96-well plate. Raji (CD19+) and K562 (CD19−) tumor cell lines were used to stimulate CD19-specific TAC-T cells, whereas SK-OV-3 (HER2+) and LOX-IMVI (HER2−) cell lines were used for HER2-specific TAC- and CAR-T cells.

BD GolgiPlug protein transport inhibitor (BD Biosciences) was added at the start of stimulation per manufacturer's instruction. After stimulation, cells were stained for surface markers as above. BD Cytofix/Cytoperm fixation and permeabilization kit (BD Biosciences) was used to permit intracellular cytokine staining and cells were stained directly for TNFα, IFNγ, and IL-2 expression. Flow cytometry was conducted as above and data was analyzed using FlowJo (FlowJo, LLC). Data for percent of T cells cytokine-positive was calculated as: cytokine TAC/CAR [%] – vector control T cell [%].

**In vitro cytotoxicity luminescence assay.** To evaluate cytotoxicity, $5 \times 10^4$ luciferase engineered cells (NALM-6, SK-OV-3, LOX-IMVI, OVCAR-3) were co-cultured with T cells in a white flat bottom 96-well plate (Corning) at indicated effector:target for 6 or 24 h at 37 °C. After co-culture, 0.15 mg/mL D-Luciferin (Perkin Elmer, Waltham, MA) was added per well and luminescence was measured using a i3 SpectraMax (Molecular Devices, Sunnyvale, CA) across all wavelengths. Tumor cell viability was calculated as: ((Emission − Background)/(Tumor cell alone − Background)) × 100%. Each condition was tested in duplicate or triplicate.

**In vitro cytotoxicity colorimetric assay.** Adherent tumor cell lines LOX-IMVI and OVCAR-3 were used to evaluate cytotoxicity of HER2-CAR- and TAC-T cells. Tumor cells were plated at $1.25 \times 10^4$ (LOX-IMVI) or $2.5 \times 10^4$ (OVCAR-3) cells/well overnight in a flat-bottom 96-well plate. T cells were evaluated at effector: target ratios of 0.25:1 to 8:1 and co-cultured for 6 h at 37 °C. After co-culture, T cells were washed off and tumor cell viability was determined using a 10% solution of AlamarBlue cell viability reagent (Life Technologies) per manufacturer's instructions. Color change, indicative of live cells, was measured by fluorescence ($\lambda_{excitation}$ 530 nm, $\lambda_{emission}$ 595 nm) on a Safire plate reader (Tecan, Maennendorf, Switzerland). Tumor cell viability was calculated as: ((Emission − Background)/(Tumor cell alone − Background)) × 100%. Each condition was tested in triplicate.

**Adoptive transfer and in vivo monitoring**. The McMaster Animal Research Ethics Board approved all murine experiments. Five-week-old female NOD. Cg-Rag1tm1MomIl2rgtm1Wjl/SzJ (NRG) mice were purchased from The Jackson Laboratory (Bar Harbor, ME) (Stock #007799), or bred in-house. Mice (6–12-weeks-old) were implanted with $2.5 \times 10^6$ OVCAR-3 cells subcutaneously (s.c.) into the right hind flank. After 35–42 days of tumor growth, mice were optimized into treatment groups based on tumor volume[50]. Engineered T cells were infused intravenously (i.v.) (deemed adoptive cell transfer (ACT)) through the tail vein as two doses delivered 48 h apart (T cells were d14 and d16 in culture on respective treatment days; doses as specified in figure legends represent the total sum of effective (transduced receptor positive) T cells received/mouse). Tumor volume was measured by caliper (Mitutoyo Canada Inc., Toronto, ON) every 2–3 days post-ACT and calculated as $L \times W \times H$; % change in tumor volume was calculated as ((current volume ($mm^3$) − pre-ACT volume ($mm^3$))/pre-ACT volume ($mm^3$)) ×100. A tumor volume endpoint of ≥2000 $mm^3$ was adhered to. Mice were weighed (OHAUS Corporation, Parsippany, NJ) every 1–3 days post-ACT; % change in weight was calculated as ((current weight (g)−pre-ACT weight (g))/pre-ACT weight (g)) × 100. Early termination criteria included moribundity or two consecutive measurements of ≥ 20% weight loss, despite supportive therapy (heat, subcutaneous fluids). Or 7–11-week-old male NRG mice were injected with $0.5 \times 10^6$ NALM6-effLuc cells intravenously. Two doses of engineered T cells were administered as above after 3 days of tumor growth. Tumor burden was monitored through bioluminescent imaging. Briefly, 10 µL/g of a 15 mg/mL D-Luciferin solution (Perkin Elmer; Waltham, MA) is injected intraperitoneally 14 min prior to dorsal and ventral imaging using an IVIS Spectrum (Caliper Life Sciences; Waltham, MA). Images were analyzed using Living Image Software v4.2 for MacOSX (Perkin Elmer) and dorsal and ventral radiance was summed. Termination criteria included moribundity or hind limb paralysis. In all cases animal treatment strictly adhered to McMaster Animal Research Ethics Board instructions and guidelines.

**Histology**. Tissues were prepared for veterinary necropsy via whole body formalin perfusion as described previously[51]. After fixation in 10% neutral buffered formalin, tissues were paraffin-embedded, sectioned and stained using hematoxylin and eosin (H&E) or immunohistochemistry (IHC) for expression of human CD3 (Abcam Inc., Toronto, ON, cat#: ab16669) (conducted using the Leica BOND RX (Leica Biosystems Inc., Concord, ON)). Aforementioned histology services were performed by the Core Histology Facility at the McMaster Immunology Research Centre. Opal multiplex IHC was performed by the Molecular and Cellular Immunology Core at the British Columbia Cancer Agency's Deeley Research Centre. In short, formalin-fixed paraffin-embedded tissue sections were stained with anti-CD4 (Abcam, cat#: ab133616) detected with Opal 520 (PerkinElmer), anti-CD8 (Spring Biosciences, cat# M3162) detected with Opal 650 (PerkinElmer), anti-HER2 (polyclonal, Cell Signaling Technology, cat#: 2242) detected with Opal 570 (PerkinElmer), anti-pan-CK Sigma Aldrich, cat#: C1801 detected with Opal 690 (PerkinElmer), anti-Ki67 (Spring Biosciences, cat#: M3062) detected with Opal 620 (PerkinElmer), and DAPI (PerkinElmer). Multispectral images (20× magnification, three fields per tumor and three fields containing perivascular sites per lung) were collected using the PerkinElmer Vectra system. Quantification was performed using inform Advanced Image Analysis Software (PerkinElmer). Blinded pathologic assessment of H&E and CD3 IHC slides was performed by a veterinary pathologist (J.M.K., McMaster University). Blinded CD3 IHC scoring was performed by a pathology resident (I.B., McMaster University)[52].

**Serum cytokine analysis**. Prior to necropsy, mice were underwent a non-terminal retro-orbital bleed. Serum was isolated using CapiJect capillary blood collection serum tubes according to manufacturer instructions (Terumo Medical Corporation, Somerset, NJ, Cat No. T-MG). Quantification of 13 human cytokines and chemokines (cat#: HDF31) or 31 murine cytokines and chemokines (cat#: MD31) was performed in a multiplex assay by Eve Technologies (Eve Technologies Corporation, Calgary, AB) using the Bio-Plex 200 system and MILLIPLEX assay kits from Millipore. The assay sensitivities of these markers ranged from 0.1–9.5 pg/mL (human) and 0.1–33.3 pg/mL (murine); individual analyte values can be found through the Eve Technologies website.

**Cell lines**. Human tumor cell lines SK-OV-3, LOX-IMVI, and OVCAR-3 originating from the NCI-60 panel (kind gift from Karen Mossman, McMaster University, Hamilton, ON) were cultured in RPMI 1640 (Gibco) supplemented with 8.7% heat-inactivated fetal bovine serum (Gibco) 1.75 mM L-glutamine (BioShop, Burlington, ON), 8.7 mM HEPES (Roche Diagnostics, Laval, QC), 87 U/mL penicillin + 87 µg/mL streptomycin (Gibco), and 48 nM β-mercaptoethanol (Gibco). Prior to use OVCAR-3 cells, were passaged in vivo. In brief, OVCAR-3 cells were injected s.c. into the hind flank of an NRG mouse and allowed to grow for 72 days prior to harvest, digested with a mixture of collagenase type I (Gibco), DNase I (Roche), and hyaluronidase (MP Biomedicals LLC, Solon, OH), and the resulting cell product was expanded ex vivo. The Raji, NALM-6, and K562 cell lines were obtained from the American Type Culture Collection (ATCC, Manassas, VA), Deutsche Sammlung von Mikroorganismen und Zellkulturen GmbH (Braunschweig, Germany), and kindly provided by Carl June (University of Pennsylvania,

Philadelphia, PA), respectively, and were cultured in RPMI 1640 (Gibco) supplemented with 8.7% heat-inactivated fetal bovine serum (FBS), 1.75 mM L-glutamine, 8.7 mM HEPES, 0.87 mM sodium pyruvate (Sigma-Aldrich Canada Co.), 0.87 × non-essential amino acids (Gibco), 48 µM β-mercaptoethanol, and 87 U/mL penicillin + 87 µg/mL streptomycin. NALM6-effLuc, SK-OV-03-effLuc, K562-effLuc, and LOX-IMVI-effLuc cells were generated by transducing tumor cells by lentivirus encoding enhanced firefly luciferase[53] and a puromycin selection marker. effLuc+ cells were selected for by supplementing culture medium with 2–8 µg/mL puromycin (InvivoGen, San Diego, CA). HEK293T cells were cultured in DMEM (Gibco) with 8.7% heat-inactivated FBS, 8.7 mM HEPES, 1.75 mM L-glutamine, 87 U/mL penicillin + 87 µg/mL streptomycin or 0.1 mg/mL normocin (InvivoGen, San Diego, CA). All cell lines were cultured under ambient atmosphere adjusted to 5% $CO_2$ and 37 °C, and confirmed mycoplasma-negative by MycoAlert mycoplasma detection kit (Lonza Inc, Basel, Switzerland). LOX-IMVI-mHER2 cells were generated by transduction of parental LOX-IMVI cells with a murine HER2 encoding lentivirus (GeneCopoeia, Cat. No.: LPP-Mm02366-Lv151-050) at an MOI of 20 and antibiotic selection with G418 (Gibco) at 500 µg/mL, according to the manufacturer's directions.

**Statistical analysis**. Multiple $t$ tests, using the Holm-Sidak method, were used to compare data between two groups. Results were prepared using Prism 6 Software (GraphPad, La Jolla, CA). A significance interval of 95% was used; NS = not significant. Power calculations were performed using G*Power[54]. Significance of survival curves was determined via the Log Rank test in Prism 6 Software (GraphPad, La Jolla, CA). To determine statistical significance of temperature, weight or tumor growth curves that appear similar, appropriate model curve fitting algorithms were used. Tumor growth was fitted to an exponential growth function ($Y = Y0*exp(k*X)$), while weight and temperature could be best fitted to a liner function ($Y = YIntercept + Slope*X$). Key parameters (exponential growth (k), linear (Slope)) determined via curve fitting were then tested for statistical significance. Curves that could not be fit by functions of the same order are considered significantly different.

**Antibodies and recombinant proteins**. Flow cytometry antibodies used: CD4-AF700 (eBioscience; cat#: 56-0048-82); CD4-Pacific Blue (BD Pharmingen; cat#: 558116); CD8-AF700 (eBioscience; cat#: 56-0086-82); CD8-PerCP-Cy5.5 (eBioscience; cat#: 45-0088-42); LNGFR-BV421 (BD Pharmingen; cat#: 562582); LNGFR-VioBright FITC (Miltenyi Biotec; cat#: 130-104-893); Human IgG (Fcγ)-PE (Jackson ImmunoResearch; cat#: 109-115-098); IFNγ-APC (BD Pharmingen; cat#: 554702); IL-2-PE (BD Pharmingen; cat#: 554566); TCR αβ-FITC (BD Pharmingen; cat#: 555547); TNFα-PE-Cy7 (BD Pharmingen; cat#: 557647); TNFα-FITC (BD Pharmingen; cat#: 554512); rhErbB2/Fc Chimera (R&D Systems; cat# 1129-ER); Protein L-Biotin (Thermo Fisher Scientific; cat#: 29997); Streptavidin-APC (BD Pharmingen; cat#: 554067); CD27-APC-H7 (BD Pharmingen; cat#: 560222); CD28-PE (BD Pharmingen; cat#: 555729); CD45RA-ECD (Beckman Coulter; cat#: IM2711U); CD62L-APC (BD Pharmingen; cat#: 559772); CCR7-PE-Cy7 (BD Pharmingen; cat#: 557648); PD-1-BV421 (BD Horizon; cat#: 562516); TIM-3-BV785 (Biolegend; cat#: 340031); LAG-3-AF647 (BD Pharmingen; cat#: 565716). IHC antibodies used: CD3 (Abcam Inc.; cat#: ab16669), CD4 (Abcam Inc.; cat#: ab133616), CD8 (Spring Biosciences, Pleasanton, CA; cat#: M3162), HER2 (Cell Signaling Technology, Danvers, MA; cat#: 2242), pan-CK (Sigma Aldrich; cat#: C1801), Ki67 (Spring Biosciences; cat#: M3062), and DAPI, Opal 520, Opal 650, Opal 570, Opal 690, and Opal 620 (Opal 7-ColorfIHC kit; Perkin Elmer; cat# NEL797001KT).

**Data availability**. The data that support the findings of this study are available from the corresponding author upon reasonable request.

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

## Acknowledgements

We gratefully acknowledge research support from the Terry Fox Foundation, Samuel Family Foundation, and Triumvira Immunologics. J.A.H. was supported by a fellowship from the Canadian Cancer Society.

## Author contributions

C.W.H., J.A.H., K.A.M., V.W.C.L., and J.L.B. conceived of various aspects of these studies and designed experiments. C.W.H. invented the TAC receptor architecture. G.F.D. made contributions to receptor design. C.W.H., J.A.H., K.A.M., V.W.C.L., A.A., K.B., L.N., D.L.H., and B.B. performed in vitro TAC assays. J.A.H. performed in vitro CAR assays. V.W.C.L. performed phenotypic analysis of checkpoint receptors and memory markers. J.A.H., K.B., and C.A. performed in vivo experiments in the HER2 model. C.A. performed in vivo experiments in the CD19 model. J.M.K. and I.B. performed pathological analysis and scoring of murine tissues, respectively. H.D. and K.M. performed multicolor IHC as designed by B.H.N., K.M., and J.L.B. C.W.H., J.A.H., K.A.M., V.W.C.L., A.A., K.B., L.N., D.L.H., and J.L.B. prepared the manuscript (figure design, writing, editing). All authors reviewed the final manuscript prior to submission.

**Additional information**

**Competing interests:** Christopher W. Helsen, Joanne A. Hammill, Kenneth A. Mwawasi, Arya Afsahi, Galina F. Denisova, and Jonathan L. Bramson hold shares in Triumvira Immunologics. Christopher W. Helsen and Jonathan L. Bramson are founding scientists of Triumvira Immunologics. All other authors declare no competing interests.

