## [Peer Review File · Nature Communications]

Reviewers' comments:

Reviewer #1 (Cancer immunotherapy, CAR T cells)(Remarks to the Author):

Authors have developed novel bispecific T cells based on recognition domains comprised of scFv or DARPins, and anti-CD3epsilon linked to a CD4 or CD8alpha coreceptor. The TAC has increased antitumor efficacy in her2: SKOV3 and NALM6 tumor models, demonstrating that it is robust. The DARPIn is worrisome in that it causes lethal off-target effects in mice. Her2 DARPIn:28:zeta CAR T responded to an unidentified antigen in the lungs and heart driving a lethal systemic inflammatory reaction. Surprisingly, HER2-TAC-T cells with the same DARPIn had antitumor effects but did not show evidence of activation or pathology within the lung and heart at the dose tested. Some issues to address:

1. In Fig. 3. authors show that TAC cells have fewer features of exhaustion and autoactivation than CD28-based CAR T. Previous studies (refs 27 and 41) have shown 4-1BB based CAR T have less autoactivation than CD28-based CAR T. Authors need to compare TAC to 4-1BB based CAR, which are now used commercially for leukemia.
2. In Fig 4, the TAC cells show promising antitumor effects. However, in the treated animals, only 15 days of follow up is shown. Authors need to show longer term observation to determine duration and safety of the of tumor response.
3. In Fig 5, authors show a surprising cytokine release syndrome in mice given her2:28:z CAR T cells. Authors need to show cytokine levels in the serum of these mice. In addition, authors need to determine if this is dependent on antigen recognition in the tumor. What happens if mice are treated with her2:28:z CAR T cells that are tumor free? It is possible that the DARPIn is reactive to a mouse target? Authors have not shown experiments with mice that are tumor free after administration of her2 TAC T cells; this should be done at a variety of doses.

Reviewer #2 (Cancer therapy, T cell therapy, DC therapy)(Remarks to the Author):

This is paper titled "A novel chimeric T cell receptor that delivers robust anti-tumor activity and low off-tumor" shows that TCR based CAR,s called TAC, in a solid tumor model demonstrated that TAC-T cells outperformed CD28-based CAR-T cells. This finding is important and could advance the field of ACT therapy.

However, there is a mix of minor and major concerns regarding the MS, below:

1. Lines 124-125, and Figure 4: Why did you only graph these curves to day 15, when you say the experiment went out to 60 days? It seems a bit dishonest to not include the rest of the data, when it could easily be graphed. Your statement that "In all cases (n=11), long-term tumor control was observed, with mice surviving to the experimental endpoint (~60 days post-ACT, data not shown)" is vague and needs to be expanded on with actual data (perhaps survival curves).
2. Figure 5: In my opinion, survival curve analysis with statistics would also be appropriate here, as well as some statistical analysis comparing the tumor curves and observed weight loss between the treatments (here and in the text).
3. Lines 147-148: It is not fair to state that weight loss is a measure of toxicity alone. To be rigorous, other toxicity assay's should be performed.
4. How do TACs compare to naturally arising TILs (tumor infiltrating lymphocytes expanded from a solid tumor)? It is possible to generate TIL or to engineer a TCR into the T cell. How is TAC and advantage of these types of thearpies, which have shown to be very promising against solid tumors like melanoma and sarcoma at the NCI in Bethesda MD.

RESPONSE TO REVIEWERS' QUERIES (Reviewers' queries in black type and authors' response in blue type)

Reviewer #1 (Cancer immunotherapy, CAR T cells)(Remarks to the Author):

Authors have developed novel bispecific T cells based on recognition domains comprised of scFv or DARPins, and anti-CD3epsilon linked to a CD4 or CD8alpha coreceptor. The TAC has increased antitumor efficacy in her2: SKOV3 and NALM6 tumor models, demonstrating that it is robust. The DARPIn is worrisome in that it causes lethal off-target effects in mice. Her2 DARPIn:28:zeta CAR T responded to an unidentified antigen in the lungs and heart driving a lethal systemic inflammatory reaction. Surprisingly, HER2-TAC-T cells with the same DARPIn had antitumor effects but did not show evidence of activation or pathology within the lung and heart at the dose tested. Some issues to address:

1. In Fig. 3, authors show that TAC cells have fewer features of exhaustion and autoactivation than CD28-based CAR T. Previous studies (refs 27 and 41) have shown 4-1BB based CAR T have less autoactivation than CD28-based CAR T. Authors need to compare TAC to 4-1BB based CAR, which are now used commercially for leukemia.

We have repeated these experiments and compared T cells expressing 2nd generation CARs with either CD28 or 4-1BB cytoplasmic domains to TAC T cells. Our experiments that T cells engineered with either CD28- and 4-1BB-based CARs display elevated expression of checkpoint receptors and diminished frequencies of cells with a naïve

phenotype (data are shown in Figure 3 and Supplementary Figure 5) indicating that both CARs display features of auto-activation. Consistent with ref 27, we observed that 4-1BBz HER2-CAR T cells exhibited reduced expression of PD-1 and TIM-3 in CD4+ T cells compared to CD28z HER2 CAR-T cells. In these repeat experiments, TAC T cells continue to show a lack auto-activation and display phenotypic characteristics that were highly similar to T cells engineered with a control vector.

2. In Fig 4, the TAC cells show promising antitumor effects. However, in the treated animals, only 15 days of follow up is shown. Authors need to show longer term observation to determine duration and safety of the of tumor response.

We agree with the reviewer and we have extended all data points in Figure 4 to 60 days.

In Figure 4B, the mice treated with TAC T cells directed against CD19, (a negative control for the OVCAR model) were only followed for 40 days. During this time, we observed no statistical difference in tumor growth relative to control and the mice were terminated. As such, the data for this particular group only extends to 40 days.

In the original data set for the CD19-TAC, we did not have tumor measurements over 60 days because the imaging device broke down. We therefore, repeated the experiment with the repaired instrument and the data are shown in Figure 4D.

3. In Fig 5, authors show a surprising cytokine release syndrome in mice given her2:28:z CAR T cells. Authors need to show cytokine levels in the serum of these mice.

Serum cytokines in tumor-bearing mice following treatment with CAR-T cells, TAC-T cells or control T cells are shown in Figure 6B.

In addition, authors need to determine if this is dependent on antigen recognition in the tumor. What happens if mice are treated with her2:28:z CAR T cells that are tumor free? It is possible that the DARPin is reactive to a mouse target?

We have performed experiments in tumor-free mice and determined that toxicity is observed in the absence of tumor. In some cases, toxicity is exacerbated in the presence of the tumor, indicating both on-tumor and off-tumor toxicities, a situation which is likely to occur in the clinical application of engineered T cells against solid tumors. The DARPin is reactive against a murine target, however the target has not been defined at this time. We have included this information at the end of the Results section (lines 217 – 219 of the revised manuscript).

Authors have not shown experiments with mice that are tumor free after administration of her2 TAC T cells; this should be done at a variety of doses.

We have not observed any toxicities in tumor-bearing mice following treatment with TAC T cells. Since tumor-free mice carry the same host antigens as tumor-bearing mice, we do not believe that additional studies with TAC T cells in tumor-free mice will offer new information that will impact the interpretation of the data in this manuscript.

Reviewer #2 (Cancer therapy, T cell therapy, DC therapy)(Remarks to the Author):

This is paper titled "A novel chimeric T cell receptor that delivers robust anti-tumor activity and low off-tumor" shows that TCR based CAR,s called TAC, in a solid tumor model demonstrated that TAC-T cells outperformed CD28-based CAR-T cells. This finding is important and could advance the field of ACT therapy. However, there is a mix of minor and major concerns regarding the MS, below:

1. Lines 124-125, and Figure 4: Why did you only graph these curves to day 15, when you say the experiment went out to 60 days? It seems a bit dishonest to not include the rest of the data, when it could easily be graphed. Your statement that "In all cases (n=11), long-term tumor control was observed, with mice surviving to the experimental endpoint (~60 days post-ACT, data not shown)" is vague and needs to be expanded on with actual data (perhaps survival curves).

We agree with the reviewer and have extended the data to 60 days as described in our response to query #2 from Reviewer #1.

2. Figure 5: In my opinion, survival curve analysis with statistics would also be appropriate here, as well as some statistical analysis comparing the tumor curves and observed weight loss between the treatments (here and in the text).

We have conducted statistical analysis of all the tumor growth curves in Figure 5 and added a survival curve in Supplementary Figure 6E. The outcomes of the analysis have been added to the Results (lines 160 – 166) The statistical analysis supports our conclusions that only the TAC and 2nd Gen CAR have a significant impact on tumor growth and that only the 2nd Gen CAR has a significant impact on loss in body weight. The details of the statistical analysis have been added to the Materials and Methods.

3. Lines 147-148: It is not fair to state that weight loss is a measure of toxicity alone. To be rigorous, other toxicity assay's should be performed.

We evaluated several signs of toxicity, including: body temperature, breathing and grooming. We have included changes in body temperature as a second quantitative measure of toxicity in Supplementary Figure 7.

4. How do TACs compare to naturally arising TILs (tumor infiltrating lymphocytes expanded from a solid tumor)? It is possible to generate TIL or to engineer a TCR into the T cell. How is TAC and advantage of these types of therapies, which have shown to be very promising against solid tumors like melanoma and sarcoma at the NCI in Bethesda MD.

We agree that TILs and TCR-engineered T cells have shown promising results in the clinic. Antigen recognition is the key difference between TAC-engineered T cells and TILs/TCR-engineered T cells. Whereas both TILs and TCR-engineered T cells recognize peptide/MHC complexes, the TAC mediates direct antigen recognition without the need for antigen processing or MHC presentation. Thus, the key advantage to the TAC is the ability to use a single receptor across all patients without the need to employ a receptor that is specific for the patient's HLA. Further, loss of MHC, a common feature in solid tumors, will impair recognition of TILs/TCR-engineered T cells but should not impact the ability of TAC-T cells to engage the target. Ultimately, further investigation will be required to define the therapeutic advantages of each modalities, but such investigations are beyond the scope of the current manuscript.

REVIEWERS' COMMENTS:

Reviewer #1 (Remarks to the Author):

no further comments. Authors have addressed previous concerns.

Reviewer #2 (Remarks to the Author):

The authors have addressed our questions. Nice work. Accepted.